# A model-based cost-utility analysis of an automated notification system for deteriorating patients on general wards

Emily Holmes[1], Huw Lloyd Williams[1], Dyfrig Hughes[1], Elke Naujokat[2], Bernd Duller[3], Christian P. Subbe [4,5] *

1 Centre for Health Economics and Medicines Evaluation (CHEME), Bangor University, Bangor, United Kingdom, 2 Philips Medizin Systeme Boeblingen GmbH, Böblingen, Germany, 3 Perpet Production, Ulm, Germany, 4 School of Medication and Health Sciences, Bangor University, Bangor, United Kingdom, 5 Betsi Cadwaladr University Health Board, Bangor, United Kingdom

* c.subbe@bangor.ac.uk

**Data Availability Statement:** All relevant data are within the paper and its Supporting Information files.

## Abstract

### Background

Delayed response to clinical deterioration of hospital inpatients is common. Deployment of an electronic automated advisory vital signs monitoring and notification system to signal clinical deterioration is associated with significant improvements in clinical outcomes but there is no evidence on the cost-effectiveness compared with routine monitoring, in the National Health Service (NHS) in the United Kingdom (UK).

### Methods

A decision analytic model was developed to estimate the cost-effectiveness of an electronic automated advisory notification system versus standard care, in adults admitted to a district general hospital. Analyses considered: (1) the cost-effectiveness of the technology based on secondary analysis of patient level data of 3787 inpatients in a before-and-after study; and (2) the cost-utility (cost per quality-adjusted life-year (QALY)) over a lifetime horizon, extrapolated using published data. Analysis was conducted from the perspective of the NHS. Uncertainty in the model was assessed using a range of sensitivity analyses.

### Results

The study population had a mean age of 68 years, 48% male, with a median inpatient stay of 6 days. Expected life expectancy at discharge was assumed to be 17.74 years. (1) Cost-effectiveness analysis: The automated notification system was more effective (-0.027 reduction in mean events per patient) and provided a cost saving of -£12.17 (-182.07 to 154.80) per patient admission. (2) Cost-utility analysis: Over a lifetime horizon the auto-mated notification system was dominant, demonstrating a positive incremental QALY gain (0.0287 QALYs, equivalent to ~10 days of perfect health) and a cost saving of £55.35. At a threshold of £20,000 per QALY, the probability of automated monitoring being cost-effective in the NHS was 81%. Increased use of cableless sensors may reduce cost-savings,

**Funding:** Yes - the study was funded by Philips Healthcare. Bangor University received funding from Betsi Cadwaladr University Health Board, Wales, UK to complete this research, via a Research Service subcontract of "Specific Research Plan for Economic Evaluation of Philips IntelliVue Guardian Solution (IGS)" incorporated into the Observational Field Test/Post Market Study Agreement between Philips Medizin Systeme Böblingen GmbH and Betsi Cadwaladr University Local Health Board.

**Competing interests:** Christian P Subbe has acted as principal investigator for clinical studies sponsored by Philips Healthcare, has served as an invited speaker for meetings hosted by Philips Healthcare and is a member of two advisory boards for Philips Healthcare. Elke Naujokat is an employee of Philips Medizin Systeme Böblingen GmbH. Bernd Duller is a consultant for Philips Medizin Systeme Böblingen GmbH.

however, the intervention remains cost-effective at 100% usage (ICER: £3,107/QALY). Stratified cost-effectiveness analysis by age, National Early Warning Score (NEWS) on admission, and primary diagnosis indicated the automated notification system was cost-effective for most strategies and that use representative of the patient population studied was the most cost-saving strategy.

## Conclusion

Automated notification system for adult patients admitted to general wards appears to be a cost-effective use in the NHS; adopting this technology could be good use of scarce resources with significance for patient safety.

## Introduction

### Clinical background

Deterioration of patients on general hospital wards often goes unnoticed for prolonged periods of time [1]. This delay can result in otherwise preventable cardiopulmonary arrest and admission to the intensive care unit (ICU) [2,3] even though, in most cases, measurable changes in vital signs [4] could identify patients at risk. Such delayed or absent response to deterioration has been labelled as "failure to rescue" [5]. To decrease the incidence and consequences of such failure to rescue, many hospitals have introduced rapid response systems (RRSs) [6] consisting of an afferent limb based on monitoring of vital signs that triggers activation of the efferent limb, individuals or teams with training in the management of critical illness. Even in hospitals with an established RRS, failure-to-rescue events occur [7–9], mostly related to problems with the afferent (monitoring, identification, and rapid response team (RRT) activation) component of the RRS. All these failings have in common the dependence on individual bedside staff to raise the alarm.

In contrast to human-based response, industrial high-reliability systems rely on redundancy to ensure that failure of a single part does not result in system failure [10,11]. When this approach is applied to monitoring in health care, systems with automated notification can be deployed to notify remote and senior healthcare professionals or RRTs who are not at the bedside to respond to deterioration [12,13]. The National Early Warning Score is a score that summarises abnormalities in vital signs such as blood pressure, heart rate, temperature through a point system ranging from zero (all parameters normal) to 20 (all parameters maximally abnormal). Deterioration can be defined as a National Early Warning Score (NEWS) of 6 or more [14]. A score of 6 leads to the activation of a practitioner with critical care skills. The notification aims to prevent further deterioration to a degree that results in the need for admission to Intensive Care, death, or cardio-pulmonary arrest. This approach can be supplemented with continuous monitoring of selected vital signs such as heart rate, respiratory rate, and oxygen saturation.

A prospective before-and-after study, 'Vital Signs to Identify, Target, and Assess Level of Care Study' (short VITAL II, ClinicalTrials.gov, NCT01692847) investigated the use of conventional vital sign monitoring enhanced by automated wearable monitoring devices, automated calculation of Early Warning Scores based on vital signs, and automated notification of clinical teams triggered by pre-defined changes in vital signs in all patients admitted to two clinical areas in a district general hospital in the UK (2139 patients before (control) and 2263

after the intervention). VITAL II concluded that deployment of automated monitoring, and notification system was associated with a reduction in mortality (8 vs 6%, p = 0.042), cardiac arrests (0.7% vs 0.09%, p = 0.002) and improved mortality for those admitted to Intensive Care (45% vs 24%, p = 0.04) [15], however, there was no health economic evidence to assess the cost-effectiveness of this intervention. Economic evaluation provides a framework in which to assess the costs and effects of alternative interventions, such as automated monitoring compared to standard care.

### Aims & objectives

We aimed to inform the cost-effective use of an automated system in the National Health Service (NHS) in the United Kingdom (UK) by conducting a model-based economic evaluation, using evidence from the VITAL II study.

## Materials and methods

### Economic evaluation overview

The study design was a model-based cost-effectiveness and cost-utility analysis using secondary data, including retrospective analysis of the Vital II Research Database (RDB).

The short-term cost-effectiveness analysis (cost per event avoided) was restricted to the inpatient episode, whilst the cost-utility analysis (cost per quality-adjusted life-year (QALY)) considered the longer-term consequences of serious adverse events to extrapolate the findings to a lifetime horizon. The QALY is a single index of both survival and health-related quality of life. The evaluation was conducted from the perspective of the NHS.

A decision analytic model was developed to represent (1) use of an electronic automated advisory vital signs monitoring and notification system to signal clinical deterioration; and (2) standard care use of non-connected spot-check monitors, as is routine in the NHS (Fig 1). The model captures all events during the inpatient stay based on data obtained from the Vital II

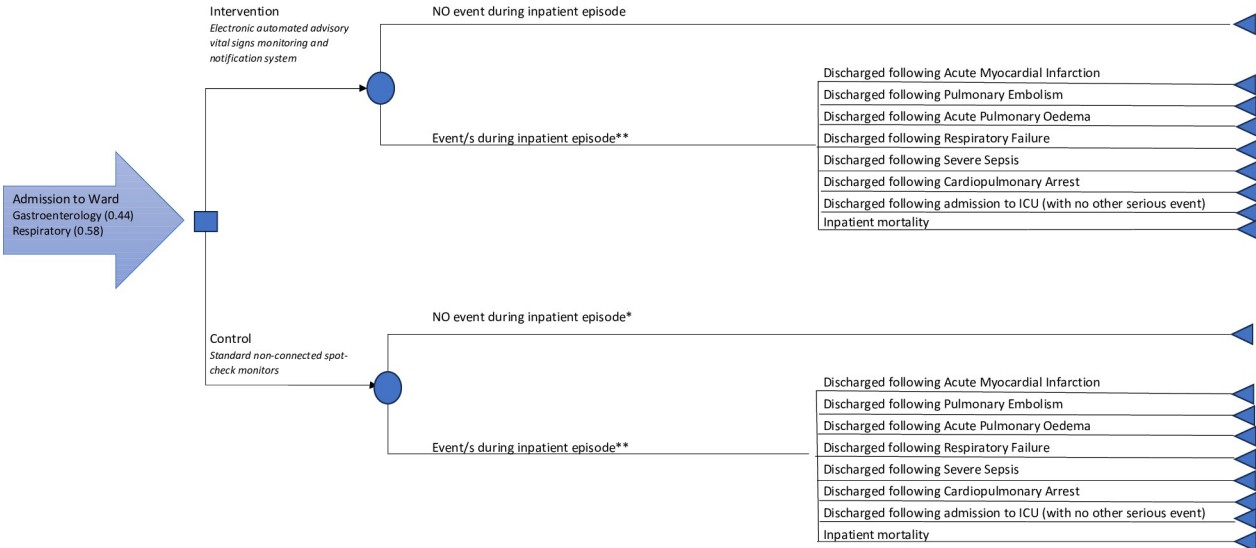

Note. ICU Intensive Care Unit . *Ward 1 Gastroenterology : Ulcerative Colitis | Crohn's disease; Ward 2 Respiratory: COPD | Pneumonia. **Serious events occurring during admission, included acute myocardial infarction, pulmonary embolism, acute pulmonary oedema, respiratory failure, severe sepsis, acute renal failure, emergency admission to the ICU, cardiopulmonary arrest, and death. Where multiple events occurred, health state at discharge defined by the principal event.

**Fig 1. Diagram of economic model.**

study. Patients were admitted to the study wards following a short period of assessment and completion of admission documentation in the Acute Medical Unit of the hospital in line with usual practice in the NHS. One of the wards specialised in Respiratory and one in Gastroenterological conditions but both wards took patients with other conditions. Once on the ward, patients in the standard care pathway were monitored in line with hospital policy, which stipulates the recording of vital signs in acutely unwell patients at least twice per day and with increasing frequency in the presence of increasing severity of illness, usually four times per day. Trained registered nurses and health care assistants obtained and recorded vital signs. Patients on the intervention pathway were monitored with an electronic automated advisory vital signs monitoring system (IntelliVue Guardian Solution (IGS) including cableless sensors and MP5SC spot-check monitors, Philips Healthcare, Boeblingen, Germany). Each spot-check monitor was used for a group of 6–8 co-located patients. During the inpatient episode 10 types of serious adverse events were collected prospectively, and these were: acute myocardial infarction, pulmonary embolism, acute pulmonary oedema, respiratory failure, stroke, severe sepsis, acute renal failure, emergency admission to the ICU, cardiopulmonary arrest, death. At discharge the model estimates lifetime costs and quality adjusted life years based on the principal serious event that occurred during the inpatient episode, or no event.

The model was parameterised using data from the VITAL II RDB (restricted to cases with complete NEWS score on admission n = 3787/4402 (86%)), and purposive reviews of the literature to obtain long-term estimates of costs and outcomes, in line with standard methodology for populating economic models [16]. Published economic evaluations were identified using UK National Institute for Health and Clinical Excellence (NICE) guidance and supplementary electronic searches of PubMed. Studies set in the UK, adopting a life-time horizon, reporting costs and QALYs for interventions/comparators that best reflected treating the condition/event in line with current practice, were selected.

The base-case model adopted a lifetime horizon to estimate the incremental cost per QALY gained, which may be used to inform decisions concerning the cost effectiveness of the intervention compared to standard care, in the UK. The analysis also reports costs per event avoided during the inpatient episode.

## Clinical parameters

**Serious adverse events/health utilities.**   Serious adverse events were obtained from the RDB. During the inpatient episode the model accounted for multiple events per patient. Health states at discharge were defined by the principal serious adverse event during inpatient episode. Where patients experienced multiple events the event with the worst health state was assumed at discharge. Each health state at discharge was assigned a Quality-Adjusted-Life-Expectancy (QALE) that was obtained from a purposive search of the literature, adjusted for the age and sex of the model population (Tables 1 and S1). The "no event" population were assigned a weighted average of chronic conditions reflecting admission to a gastroenterology ward (Crohn's Disease) or a respiratory ward (Chronic obstructive pulmonary disease (COPD) or Pneumonia).

**Resource use.**   During the inpatient episode resource use included length of stay on ward of admission (based on reason for admission and any subsequent serious adverse events), admission to ICU, use of monitoring equipment (the automated monitoring and notification system for the intervention arm, and non-connected spot-check monitors in standard care). Post-discharge resource use was not available at a patient level and is captured within life-time costs (Table 1), calculated using secondary data [external to the VITAL II clinical study].

**Unit costs.**   Unit costs associated with monitoring devices and inpatient stay were obtained from the manufacturer and the NHS sources (Table 2). The cost of the intervention

**Table 1. Cost-utility model input parameters: Principal event probabilities, lifetime costs and quality-adjusted life years.**

| Parameter | Point Estimate | Distribution[1] | References |
|---|---|---|---|
| **EVENT PROBABILITIES** | Probability | | |
| No Event_intervention | 0.9451 | Dirichlet-multinominal (3579, 24, 184) | [Footnote 2] |
| Event survive_intervention | 0.0064 | | |
| Inpatient mortality _intervention | 0.0485 | | |
| No Event_control | 0.9395 | Dirichlet-multinominal (3558, 40, 189) | [Footnote 2] |
| Event survive_control | 0.0106 | | |
| Inpatient mortality_control | 0.0499 | | |
| Non-fatal **principal event_in**tervention | | | |
| Acute Myocardial Infarction | <0.00000001 | Dirichlet-multinominal (4.9231E-13, 0.0016, 0.0011, 3.6014E-13, 3.3553E-07, 24.3621, 1.9882E-07) | [Footnote 2] |
| Pulmonary Embolism | 0.00006559 | | |
| Acute Pulmonary Oedema | 0.00004368 | | |
| Respiratory Failure | 0.00000000 | | |
| Severe Sepsis | 0.00000001 | | |
| Emergency admission to ICU | 0.99989071 | | |
| Cardiopulmonary arrest | 0.00000001 | | |
| Non-fatal **principal event__**control | | | |
| Acute Myocardial Infarction | 0.00000007 | Dirichlet-multinominal (2.9955E-06, 0.0051, 0.0010 4.9610E-06, 7.7047, 29.9749, 2.4865) | [Footnote 2] |
| Pulmonary Embolism | 0.00012726 | | |
| Acute Pulmonary Oedema | 0.00002508 | | |
| Respiratory Failure | 0.00000012 | | |
| Severe Sepsis | 0.19179070 | | |
| Emergency admission to ICU | 0.74615961 | | |
| Cardiopulmonary arrest | 0.06189716 | | |
| **INPATIENT COSTS** | Inpatient Cost | | |
| Inpatient episode cost _control | 2059.16 | 95% Central Range (1,957.03 to 2,174.21) | [Footnote 3] |
| Inpatient episode cost _intervention | 2046.99 | 95% Central Range (1,926.45 to 2,183.47) | |
| **LIFETIME COSTS** | Lifetime Cost | | |
| Ward 1_Gastroenterology | £28,694 | Gamma (25, 1147.75) | Bodger et al. (2009) [17] |
| Ward 2_Respiratory | £10,555 | Gamma (25, 422.19) | NICE (2019), (2014) |
| Acute Myocardial Infarction | £34,398 | Gamma (25, 1375.91) | NICE (2020a) |
| Acute Pulmonary Oedema | £19,198 | Gamma (25, 741.57) | Peek (2010) [18] |
| Respiratory Failure | £19,198 | Gamma (25, 767.92) | Peek (2010)[18] |
| Severe Sepsis | £45,903 | Gamma (25, 1836.14) | Soares (2012)[19] |
| Emergency admission to ICU | £19,198 | Gamma (25, 767.92) | Peek (2010)[18] |
| Cardiopulmonary arrest | £38,303 | Gamma (25, 1532.14) | Javanbakht (2022)[20] |
| **LIFETIME QALYS** | QALE[4] | | |
| Healthy population (age, sex matched) | 9.7732 | | McNamara (2023)[21] |
| Ward 1_Gastroenterology | 7.4965 | Normal (7.50, 1.50) | Bodger et al. (2009)[17] |

*(Continued)*

**Table 1.** (Continued)

| Parameter | Point Estimate | Distribution[1] | References |
|---|---|---|---|
| Ward 2_Respiratory | 7.9866 | Normal (7.99, 1.60) | |
| COPD | 4.8068 | | NICE (2019)[22] |
| Pneumonia | 9.1604 | | NICE (2014)[23] |
| Acute Myocardial Infarction | 6.0139 | Normal (6.01, 1.20) | NICE (2020)[24] |
| Pulmonary Embolism | 6.9533 | Normal (6.95, 1.39) | NICE (2020)[25] |
| Acute Pulmonary Oedema | 4.0633 | Normal (4.06, 0.81) | Peek (2010)[18] |
| Respiratory Failure | 4.0633 | Normal (4.06, 0.81) | Peek (2010)[18] |
| Severe Sepsis | 3.3345 | Normal (3.33, 0.67) | Soares (2012)[19] |
| Emergency admission to ICU | 4.0663 | Normal (4.06, 0.81) | Peek (2010)[18] |
| Cardiopulmonary arrest | 3.0013 | Normal (3.00, 0.60) | Javanbakht (2022)[20] |
| **RESOURCE USE** | Resource Use | | |
| Number of beds (n) | 54 | Fixed | VITAL II clinical study RDB n = 3787 |
| Mean length of stay (days) _intervention | 8.62 | Fixed | |
| Mean length of stay (days) _control | 8.90 | Fixed | |
| Cableless Sensor Use (rate) | 0.123 | Fixed | |
| Estimated product life (years) | 5 | Fixed | Assumption |

Note. [1]Distribution used in probabilistic sensitivity analysis: Dirichlet-multinominal (n events of N = 3787); Gamma (alpha, beta), Normal (mean, standard deviation).

[2]Estimated using mlogit to adjust for baseline differences in intervention group, age, sex, ward, base score on admission, on RDB (n = 3787).

[3]Estimated using GLM (with gamma family and log link) to adjust for baseline differences in intervention group, age sex, ward, base score on admission, on RDB (n = 3787) parameter uncertainty represented by 10,000 bootstrap replications.

[4]See S1 File for worked example.

was calculated using information provided by the manufacturer, and resource use observed in VITAL II. To calculate the mean cost of the intervention per patient, the purchase price was annualised as follows:

Mean cost of technology per patient = [(purchase price / product-life) + variable costs for 1-year] annual number of patients

Where: purchase price = fixed cost of IGS and MP5SC Monitors; variable costs = Health DOT wireless sensors, mean length of stay is days from admission to discharge; and annual number of patients = [(365/mean length of stay) * total number of beds with automated notification system enabled]. Assuming ward operates at 100% annual capacity and interest rate 0%.

The unit cost of the non-connected spot-check monitors used in standard care are understood to be included within NHS activity costs (used to cost the inpatient stay), however, on the basis that IGS would displace the cost of the spot-check monitors, a unit cost for the monitors used in the control phase of the Vital II study, was included in the analysis.

NHS Reference Costs and the National Tariff (2020/21) were used to estimate the cost of hospital stay (NHS National Cost Collection database (2021)) (S1–S4 Tables). A weighted

**Table 2. Unit costs of monitoring and inpatient stay.**

| Monitoring Device Costs *(based on technology for two wards)* | | Cost (£) |
|---|---|---|
| Intervention: IntelliVue Guardian Solution (IGS) with cableless sensors and MP5SC spot-check monitors (Philips Healthcare, Boeblingen, Germany) | | |
| Fixed costs: IGS + 12 MP5SC spot-check monitors | | £77,448.61 |
| Variable costs: Health DOT (cost per sensor)^ | | £107.50 |
| Control: Cost of spot-check monitors used in routine care at district general hospital in UK | | |
| Fixed cost: 12 Routine care spot-check monitors | | £16,800 |
| Inpatient Costs | Non-elective cost[a] | Cost per excess bed day[b] |
| Ward 1 (gastroenterology)* | £1,457 | £259 |
| Ward 2 (pulmonology)* | £1,641 | £230 |
| Acute Myocardial Infarction | £1,592 | £264 |
| Pulmonary Embolus | £1,525 | £230 |
| Acute Pulmonary Oedema | £1,543 | £230 |
| Respiratory Failure | £848 | £230 |
| Stroke | £3,609 | £257 |
| Severe Sepsis | £2,385 | £239 |
| Acute Renal Failure | £1,398 | £239 |
| ICU (bed day) | £1,620 | n/a |
| Cardiopulmonary Arrest | £1,628 | £264 |

^ The Vital II study [15] reported 12.3% of the intervention arm had at least one cableless sensor attached in the intervention phase, these represent an additional variable cost to using IGS during this phase. In the current analysis, Health DOT wireless sensors were substituted as an approximation of the costs for the cableless sensors as the latter are no longer on the market.

*Calculated as frequency weighted average of non-elective activity (currency descriptions unavailable at district general hospital excluded prior to weighting); see S1–S4 Tables for detailed activity codes / descriptions. a NHS Reference costs 2020/21. b NHS National Tariff 2020/21. See S5 Table for excess bed day trimpoints.

average of total non-elective activity was calculated, for each episode. The NHS tariff was then used to obtain trim points and costs per excess bed day for non-elective activity (S5 Table). ICU and serious adverse event activity costs were added to ward costs to provide a cost from admission to discharge/death (Table 2).

Costs incurred during the inpatient stay were not discounted due to the time horizon of less than one-year. Life-time costs and QALYs were discounted at a rate of 3.5% All costs were reported as UK pounds, price year 2020/21 for NHS costs and most recent pricing for the intervention.

**Long-term costs.** Life-time costs associated with each health state at discharge were obtained from a purposive review of published literature. As with QALE, the "no event" population were assigned a weighted average of chronic conditions. Where the event health state was associated with a higher cost than "no event" the cost of being in the event state was carried forward (all cases except pulmonary embolism). Lifetime costs were inflated to 2020/21 using the NHS Cost Inflation Index [26,27] and scaled to reflect life-expectancy of the model population (17.74-years based on age 68-years, 48% male), using published Life Expectancy Norms for the English Population accounting for age and sex [21]. Costs incurred during the inpatient episode were added to life-time costs to determine total cost over the life-time horizon.

## Analysis

Number of events were summed for each patient in the observational study and probability of event calculated using negative binomial regression to allow for baseline differences in age, gender, ward, and NEWS score on admission. Length of stay on the ward was calculated as the date of discharge, minus day of admission, minus anytime in ICU. Total hospital costs for each patient were calculated as the sum of device (automated or spot check), and inpatient stay costs (ward, ICU and serious adverse event activity costs). Hospital costs were analysed using generalized linear regression models (GLM) with gamma family and log link. Count data of events were analysed using negative binominal regression. The 95% central range for difference in events were calculated using non-parametric bootstrap analysis with 10,000 replications.

## Cost effectiveness analysis

The cost-effectiveness analysis considered the cost per event avoided and cost per life-years saved (during the inpatient episode). The Incremental Cost Effectiveness Ratio (ICER) was calculated as the incremental cost divided by the total number of events avoided or life-years gained.

**Cost per QALY.**   Total Cost and QALE data were combined to calculate the ICER. The ICER of the lifetime cost-utility analysis was calculated as follows:

$$ICER = \frac{COST_{\text{with IGS}} - COST_{\text{standard care no IGS}}}{QALE_{\text{with IGS}} - QALE_{\text{standard care no IGS}}}$$

**Base-case analysis.**   The base-case analysis assumed a monitoring device product life of 5-years and 12% cableless sensor use in the intervention arm and extrapolated to a life-time horizon.

**Sensitivity analyses.**   One-way sensitivity analysis was conducted on (1) product life from 5-year to 10-year or 15-years, (2) cableless sensors use from rates of 0% to 100%. A threshold analysis was conducted to establish the cost [and throughput] of testing at which the ICER is dominant (cost neutral/saving and more effective). Calculation of equivalent annual cost calculation based on product life of 5-years and a 3.5% discount rate / annuity factor 4.515 was also performed to assess impact on product price per patient.

**Probabilistic sensitivity analyses.**   Probabilistic sensitivity analysis was performed on the cost-utility analysis, using Monte Carlo simulation with 10,000 replications sampled from the distributions presented in Table 1. Standard deviation was assumed to be 0.2 of the mean point estimate and parameters of distributions calculated accordingly, the assumption of this was tested using scenario analysis of 0.1 and 0.4. A cost-effectiveness acceptability curve (CEAC) was constructed to illustrate the probability of testing being cost-effective at given thresholds of cost-effectiveness [28].

**Subgroup analyses.**   Subgroup analyses was conducted on clinically meaningful subgroups of (1) Age (17-74-years, 75-years +); (2) NEWS score on admission (3+, 6+); and (3) ICD 10 code of primary diagnosis (ICD 10 Diseases of respiratory system, ICD 11 Diseases of digestive system, "other" primary diagnosis i.e., not ICD 10 or 11). Patient level data were stratified into groups and model parameters were re-calculated. Secondary parameters used in the cost-utility analysis were adjusted for subgroup population age, sex, ward, and COPD/CFA status (S6 Table). To allow for comparative cost-effectiveness within and between groups the net monetary benefit (at the £20,000 per QALY threshold) and net health benefit of each strategy was calculated and plotted on the cost-effectiveness plane.

All data were analysed in Microsoft® Excel® for Microsoft 365 MSO (16.0.13801.20442) or STATA 17 and the study is reported according to the Consolidated Health Economic Evaluation Reporting Standards [29].

Research governance

The VITAL II before-and-after study was approved by the hospital human research ethics committee (Reference 12/WA/0050, Protocol number SD-05163-BBN-IGS A.2). This study recruited patients from the 5th of October 2012 to the 17th of April 2015.

The VITAL II Study Data Base (VSDB) was de-identified according to the Health Insurance Portability Act–HIPAA (full de-identification). This new fully de-identified RDB was approved by IRAS (REC reference: 21/WA/0172; IRAS project ID: 298601) and the economic evaluation was approved by Bangor University Healthcare & Medical Sciences Academic Ethics Committee (16/07/2021) and Health Care Research Wales (HCRW)(21/09/2021). Patient consent was not required. Data was accessed for research purposes on the 11[th] of October 2021. Authors of this manuscript had no access to information that could identify individual participants during or after data collection.

## Results

### Base case analyses

The study population (n = 3787) had a median age of 71 years (Inter Quartile Range (IQR): 59–81), 52% were female, just over half were admitted to the pulmonology ward (56%), and the mean NEWS value on hospital admission was 3.15 (sd = 2.82) (S7 Table). Based on (unadjusted) observed data the frequency of adverse events per patient was lower with IGS (1.15 intervention versus 1.37 control). (S8 Table).

**Short-term cost-effectiveness analysis.** The device cost for using the automated intervention was estimated to be £846 per bed per year, which equates to £19.98 per patient episode (based on 2,287 patients per year); compared to £1.52 per patient for spot-check monitors in standard care (Table 3). The total NHS cost for the hospital episode, however, was lower with the intervention (£2047 IGS, compared to £2059 control), driven by higher cost of treating events. IGS was also associated with improved health outcome (– 2.7% reduction in serious adverse events) (Table 4).

**Life-time cost-utility analysis.** Extrapolating the results from discharge to a lifetime horizon, by modelling differences in lifetime costs and QALYs, showed IGS was associated with a mean QALE of 7.37 (95% CI: 5.29 to 9.47) compared to a QALE of 7.34 (95% CI: 5.27 to 9.42) for standard care. Mean total costs over a lifetime were £19,692 (95% CI: £14,931 to £24,978) for the intervention and £19,747 (95% CI: £15,022 to £25,048) for standard care. Mean

**Table 3. Cost of intervention automated monitoring and notification and control spot-check monitoring.**

| | Intervention<br>IntelliVue Guardian Solution (IGS) with cableless sensors and MP5SC spot-check monitors (Philips Healthcare, Boeblingen, Germany) | Control<br>Cost of spot-check monitors used in routine care at district general hospital in UK |
|---|---|---|
| Total Cost (for 1-year) | £45,691.66 | £3,360.00 |
| Total cost per bed per year$ | £846.14 | £62.22 |
| Cost per patient episode $ | £19.98 | £1.52 |

Note. $Base case: 5-yrs, 54 beds, 0.12 cableless; #Base case: 5-yrs, 54 beds, 0.00 cableless using straight line depreciation. Economic equivalent annual cost calculation based on product life of 5-years and a 3.5% discount rate / annuity factor 4.515: £20.71 intervention; £1.68 control.

**Table 4. Cost effectiveness of an automated notification system for deteriorating ward patients in a district general hospital.**

| | Intervention (95% CR) | Control (95% CR) | Incremental (95% CR) |
|---|---|---|---|
| **Costs** | | | |
| Hospital Costs (£, short-term) | 2,046.99 (1,926.45 to 2,183.47) | 2,059.16 (1,957.03 to 2,174.21) | -12.17 (-182.07 to 154.80) |
| Lifetime Costs (£) | 17,644.52 (12,913.48 to 22,958.80) | 17,687.70 (12,985.16 to 22,961.88) | -43.18 (-225.16 to 163.09) |
| Total Cost | 19691.52 (14930.96 to 24977.91) | 19746.86 (15021.67 to 25048.15) | -55.35 (-309.26 to 209.39) |
| **Effectiveness (short-term)** | | | |
| Predicted count of Events (mean n events per patient) | 0.0666 (0.0543 to 0.0786) | 0.0933 (0.0743 to 0.1114) | -0.0267 (-0.0475 to -0.0064) |
| Quality-adjusted-life-expectancy (lifetime) | 7.3702 (5.2892 to 9.4685) | 7.3415 (5.2678 to 9.4200) | 0.0287 (-0.0485 to 0.1097) |

Note. CR: Central Range.

incremental QALYs was estimated to be 0.029, which is equivalent to ~10 days of perfect health; whilst mean incremental cost was estimated to be -£55.35 (Table 4).

## Results of the subgroup and sensitivity analyses

**Results of the sensitivity analyses.** The cost-effectiveness of IGS was robust to changes in product life and dominant to a cableless senor rate of 0.23. The threshold at which IGS becomes more costly is £32.06 i.e., a 60% increase in cost per patient inpatient stay (S9 Table). Economic equivalent annual cost calculation based on product life of 5-years and a 3.5% discount rate / annuity factor 4.515 made a minor adjustment to incremental cost (£0.56).

**Probabilistic sensitivity analysis.** The cost-effectiveness plane for the cost effectiveness analysis (£/events) is illustrated in Fig 2. This shows the distribution of simulations for the cost per event avoided analysis in the short term (to discharge)–the majority of simulations show a reduction in events (to the left of the y axis), with wider variation in incremental cost (above and below the x axis).

The cost-effectiveness plane for the base-case cost-utility analysis is illustrated in the Fig 3. The distribution of the simulations indicates that IGS results in high utility (health gain) but at a lower cost in 50% of simulations (south-east quadrant). The cost-effectiveness acceptability curve (CEAC) (Fig 4) indicates the probability of IGS being cost-effective is 81% at the £20,000 threshold; and, 80% at the £30,000 per QALY thresholds (upper and lower end of the UK healthcare decision making threshold for cost-effectiveness); this was robust to changes in parameter uncertainty (at the £20,000 threshold: from 79% with standard deviation 0.4 of the mean to 82% with standard deviation of 0.1 of the mean).

**Results of the subgroup analyses.** Stratified cost-effectiveness analysis indicated the automated notification system was cost-effective for all strategies, except for NEWS on admission 6 +, where the ICER was in the south-west quadrant of the cost-effectiveness plane (cost saving but less effective) and did not reach the threshold for cost-effectiveness on the UK NHS (S10 Table). Whilst automated monitoring of patients under 75-years provided the greatest net benefit and was relatively more cost-effective compared to the older subgroup; the adoption of automated monitoring remains the dominant strategy—associated with increased health gain and cost savings, over a lifetime horizon–in subgroups defined as older age, NEWS on admission less than 6, and primary ICD codes of 10 or 11 (S1–S4 Figs). The base-case (all patients) resulted in the greatest cost-saving.

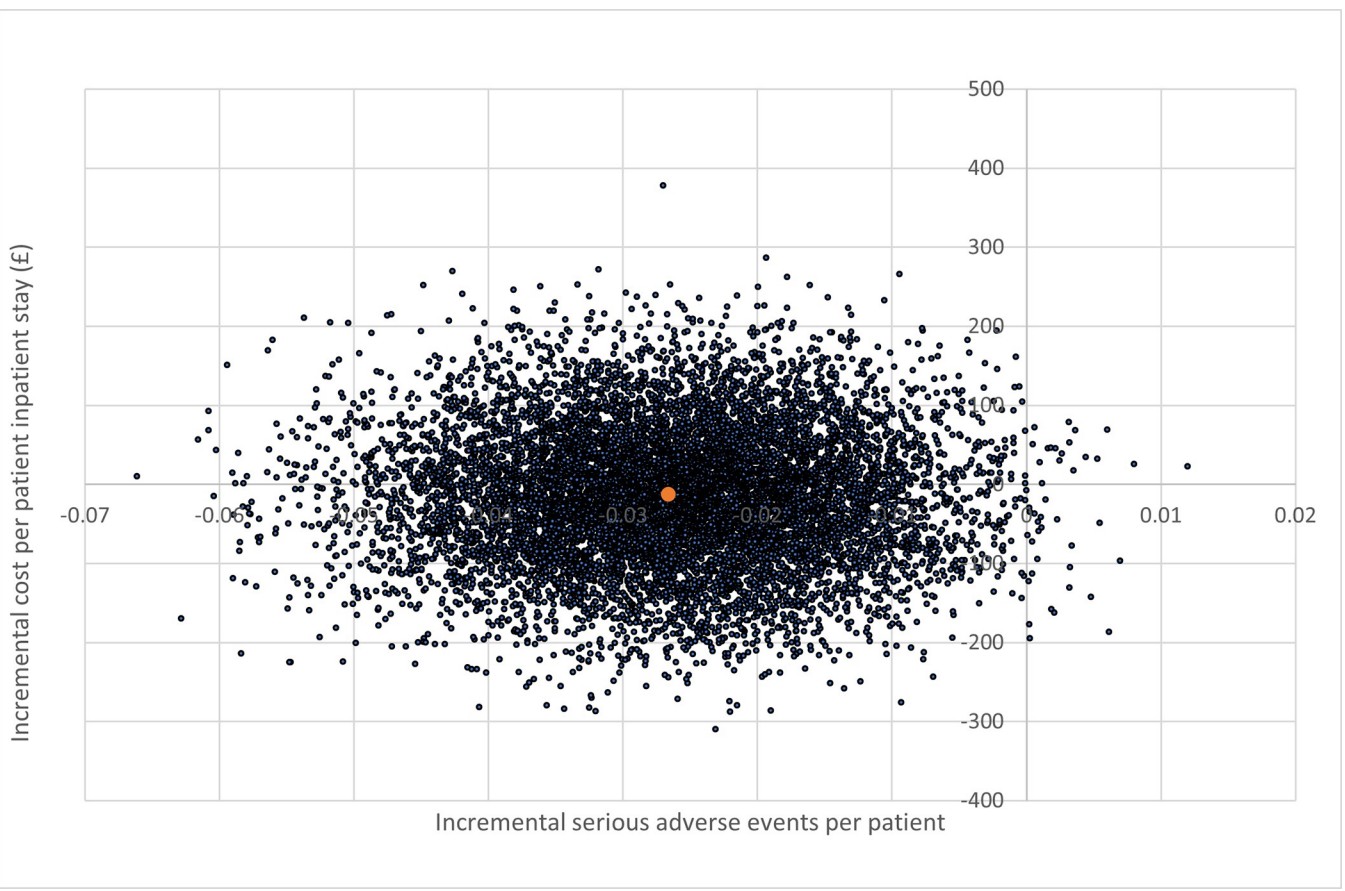

**Fig 2. Cost-effectiveness plane: Cost-effectiveness analysis £/event avoided during inpatient stay.**

## Discussion

Use of an automated notification system for deteriorating ward patients was cost-effective and associated with small costing saving in the analysis of data from a previous interventional study from the UK. Increased use of cableless sensors is associated with higher costs, however, the intervention remains cost-effective even when the rate is 100% (ICER: £3,107/QALY). Stratified cost-effectiveness analyses indicated that IGS, compared to spot-check monitors used in standard care, remains cost effective (dominant or below the ICER threshold for decision making) in all subgroups except NEWS on admission 6+.

Mohr at al. [30] conducted a retrospective analysis of implementing an early deterioration detection solution for general care in patients at a US hospital. The study used Medicare inpatient claims for a regional hospital, that reported on 445 patient admissions, majority over age 65-years and over half female. Average hospital costs per discharge were reduced by 18%, average LOS was significantly reduced–driven by a reduction in general care LOS. Complications, in-hospital mortality, and 30-day all cause readmissions were similar. We report a significant reduction in serious adverse events, and when extrapolated to a lifetime, a small improvement in QALYs. Our UK study also reports cost reductions, but of much smaller magnitude than this US study, which may in part be explained by differences in costing processes—furthermore, we do not have data on re-admission.

Vroman [31] also reported on the economics of continuous vital sign monitoring in patients after elective abdominal surgery–their retrospective analysis of clinical outcomes and

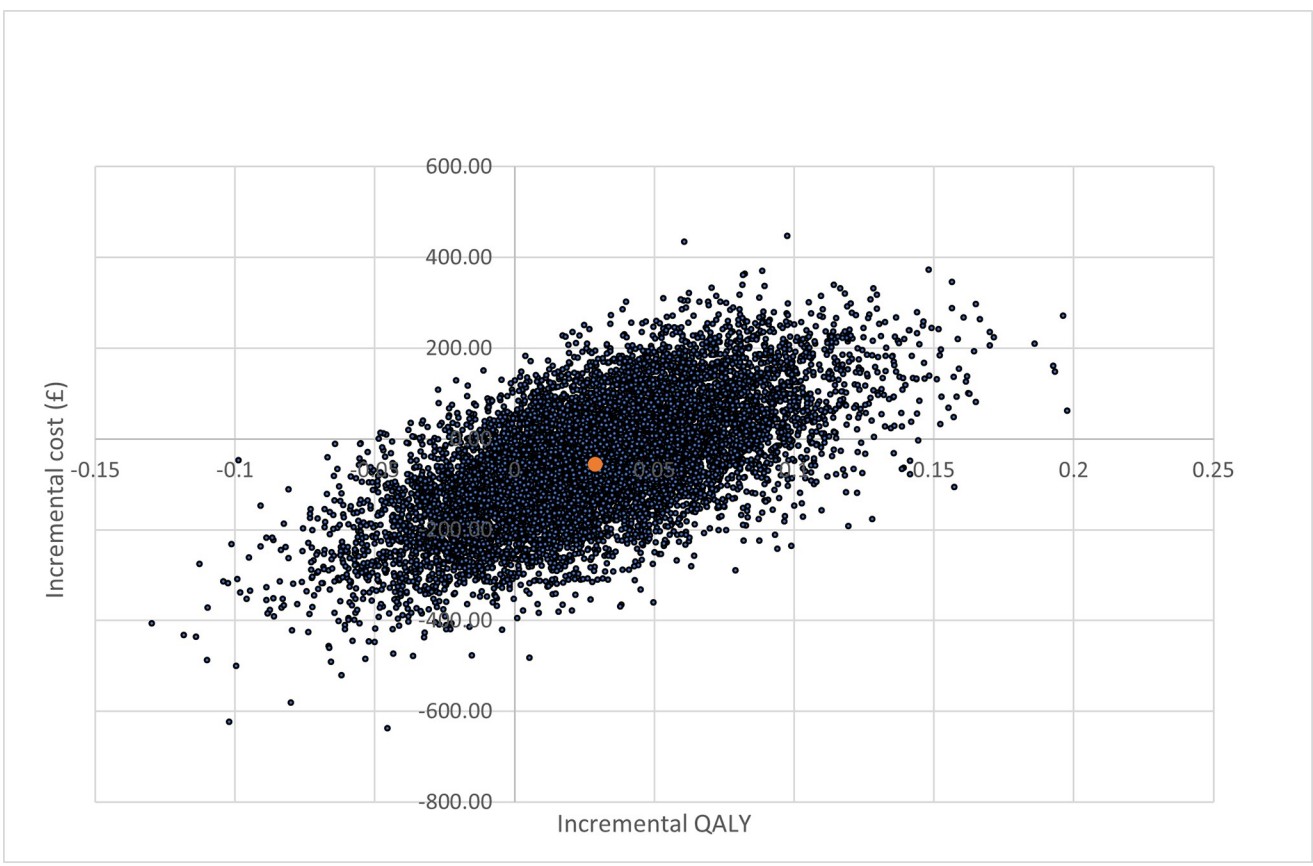

**Fig 3. Cost-effectiveness plane: Cost-utility analysis with life-time horizon.**

in-hospital costs reported less frequent ICU admissions, shorter length of stay and lower costs, in the intervention phase. The analysis was based on 855 patients in a Dutch hospital, of similar age and gender to the current UK evaluation. In this study interest was more focused on continuous monitoring with the wearable biosensor, but the findings appear comparable for the inpatient episode.

## Strengths

To our knowledge, this is the first study from the UK to model the cost-effectiveness of an electronic automated advisory vital signs monitoring and notification system. The present study used data from the VITAL II study, and therefore the probabilities in the model were based on individual patient-level data, collected, that reflected real-world situations. Furthermore, the study extrapolated beyond hospital discharge to model a lifetime horizon, to capture the full costs and outcomes potentially associated with a change in monitoring technology.

## Limitations

The analysis did not account for maintenance costs of the electronic automated advisory vital signs monitoring and notification system, or routine spot-check monitoring. It was assumed the intervention would displace existing requirements; however, it may be reasonable to estimate a 10% increase to cover training and maintenance, in which case the intervention would remain dominant. The time horizon of the cost-effectiveness model was limited to duration of inpatient stay, however, we extrapolated to a lifetime horizon to minimise time horizon bias.

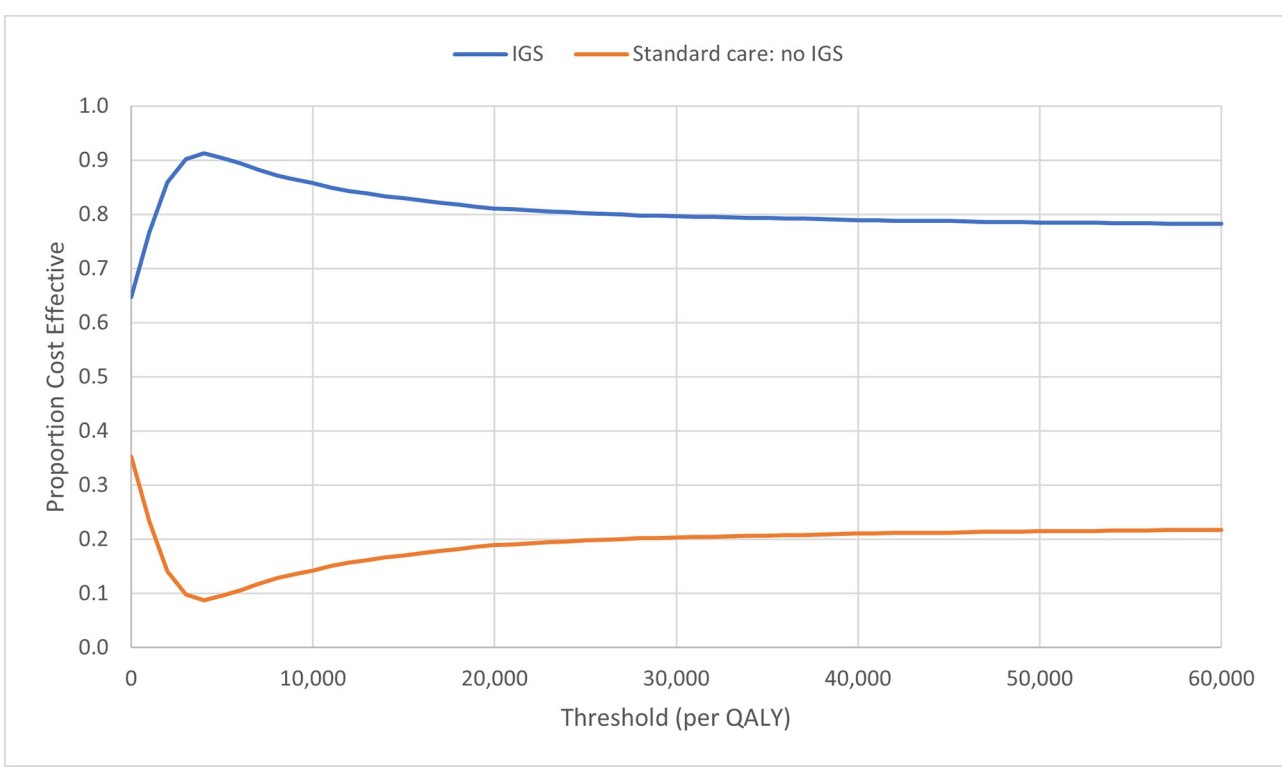

**Fig 4. Cost-effectiveness acceptability plane: Cost-utility analysis with life-time horizon.**

Whilst utility data were not collected at a patient level, we used published estimates from UK studies, that were adjusted for age and sex to match the patient population observed in the VITAL II clinical study. The analysis did not account for the opportunity cost of automated versus human monitoring, whilst this replicates policy (staffing levels are required to remain constant), time spent on monitoring represents resource that could be redistributed to other elements of care. It is also noted that the economic evaluation used a reduced sample of the before-and-after study (n = 3787/4402) and whilst adjusted probabilities used in the model are a robust reflection of available data, difference in point estimates of mortality between intervention and control of complete cases are more conservative than those reported in the effectiveness study [15], which may underestimate the cost-effectiveness of the intervention. Finally, the assumption of 100% ward capacity, may be judged to be an optimistic bound, however, it is usual practice in NHS hospitals to fill ward to capacity to create space at 'the front door' for assessment of new patients.

## Implications

This analysis highlights the cost-effectiveness of using an electronic automated advisory vital sign monitoring and notification system for patients on general wards. Based on our previous publication investment in the intervention is likely to have a significant effect on patient outcomes, while having potential cost-savings–suggests good use of scarce resources.

## Future research directions

Further research, collecting health utilities and long-term health and social care resource use is required for a more robust estimate of costs and outcomes.

The impact of automated monitoring solutions on staffing also warrants further exploration.

## Conclusion

Pragmatic use of automated monitoring in routine clinical practice for acute emergency admissions on general wards is an economically dominant strategy, where the joint distribution of costs and QALYs is associated with a positive net benefit. Adopting this technology is likely to result in both reduced costs and improved outcomes.

## Supporting information

**S1 Checklist. Consolidated Health Economic Evaluation Reporting Standards (CHEERS) checklist.**
(DOCX)

**S1 Fig. Cost-effectiveness plane for base-case and all subgroup analyses.**
(DOCX)

**S2 Fig. Cost-effectiveness plane for base-case and all subgroups by age.**
(DOCX)

**S3 Fig. Cost-effectiveness plane for base-case and all subgroups by NEWS on admission.**
(DOCX)

**S4 Fig. Cost-effectiveness plane for base-case and all subgroups by Primary ICD code.**
(DOCX)

**S1 Table. NHS reference costs ward 1 (gastroenterology) inclusion and exclusion.**
(DOCX)

**S2 Table. NHS reference costs ward 2 (pulmonology) inclusion and exclusion.**
(DOCX)

**S3 Table. NHS reference costs critical care.**
(DOCX)

**S4 Table. NHS reference costs serious events.**
(DOCX)

**S5 Table. Trimpoints used to calculate excess bed days.**
(DOCX)

**S6 Table. Baseline characteristics of subgroup model populations.**
(DOCX)

**S7 Table. Patient characteristics.**
(DOCX)

**S8 Table. Unadjusted frequency of serious adverse events and associated model probabilities.**
(DOCX)

**S9 Table. Results of sensitivity and scenario analyses.**
(DOCX)

**S10 Table. Net health benefit and net monetary benefit of alternative strategies [sub-groups].**
(DOCX)

**S1 File. Worked example of QALE calculation.**
(DOCX)

## Acknowledgments

The authors would like to thank Laura Longshaw for her tireless work and expertise in bringing the approvals for this study together. Lynne Grundy acted as sponsor for the study. The authors would like to thank the clinical teams involved in the original study for their commitment to quality and safety of care. The authors would also like to express sincere gratitude to Goran Medic from Philips who has contributed to the successful completion of this article.

## Author Contributions

**Conceptualization:** Emily Holmes, Dyfrig Hughes, Elke Naujokat, Christian P. Subbe.

**Data curation:** Bernd Duller.

**Formal analysis:** Emily Holmes, Huw Lloyd Williams, Dyfrig Hughes.

**Funding acquisition:** Elke Naujokat.

**Investigation:** Elke Naujokat, Christian P. Subbe.

**Methodology:** Emily Holmes, Huw Lloyd Williams, Dyfrig Hughes, Bernd Duller.

**Resources:** Elke Naujokat.

**Supervision:** Elke Naujokat, Christian P. Subbe.

**Writing – original draft:** Emily Holmes, Dyfrig Hughes, Bernd Duller, Christian P. Subbe.

**Writing – review & editing:** Emily Holmes, Elke Naujokat, Bernd Duller, Christian P. Subbe.

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
