## [Decision Letter · Decision Letter 0]

11 Oct 2023

PONE-D-23-27466A model-based cost-utility analysis of an automated notification system for deteriorating patients on general wardsPLOS ONE

Dear Dr. Subbe,

Thank you for submitting your manuscript to PLOS ONE. After careful consideration, we feel that it has merit but does not fully meet PLOS ONE’s publication criteria as it currently stands. Therefore, we invite you to submit a revised version of the manuscript that addresses the points raised during the review process.

We look forward to receiving your revised manuscript.

Kind regards,

Dominic Luke Thorrington, PhD

Academic Editor

PLOS ONE

Journal Requirements:

 "Yes - the study was funded by Philips Healthcare. Bangor University received funding from Betsi Cadwaladr University Health Board, Wales, UK to complete this research, via a Research Service subcontract of “Specific Research Plan for Economic Evaluation of Philips IntelliVue Guardian Solution (IGS)” incorporated into the Observational Field Test/Post Market Study Agreement between Philips Medizin Systeme Böblingen GmbH and Betsi Cadwaladr University Local Health Board."  

Additional Editor Comments:

Thank you for your submission to PLOS One and for your very interesting manuscript.

Based on my own reading of the manuscript, but more so from the comments from the second reviewer, your manuscript requires major revision before it can be published. The second reviewer has included several detailed comments that require addressing, but should improve the manuscript.

Reviewers' comments:

Reviewer's Responses to Questions

**Comments to the Author**

1. Is the manuscript technically sound, and do the data support the conclusions?

Reviewer #1: Yes

Reviewer #2: Yes

2. Has the statistical analysis been performed appropriately and rigorously? 

Reviewer #1: No

Reviewer #2: Yes

3. Have the authors made all data underlying the findings in their manuscript fully available?

Reviewer #1: No

Reviewer #2: Yes

4. Is the manuscript presented in an intelligible fashion and written in standard English?

Reviewer #1: Yes

Reviewer #2: Yes

5. Review Comments to the Author

Reviewer #1: The method is good but I do not agree with this topic. In a hospital with deteriorating patients both deployment of an electronic automated advisory vital signs and routine monitoring are needed. So it is crystal clear and does not need a study. You can not replace them but to have both. I can not accept this topic at all.

Reviewer #2: This paper presents a policy-relevant cost-effectiveness investigation utilizing peer-reviewed inputs and generally standard approaches. The analysis benefits both from a well-performed sensitivity analysis that allows for joint fluctuation in all parameters and from an array of sub-group analyses. The paper is well organized and the presentation is generally clear. Overall this is a solid analysis.

This paper does contain at least one key flaw, but one that should be fixable in a fairly straightforward way. For all its elegance in probabilistically sampling an array of input parameters for its sensitivity analysis, the fundamental costing of the intervention is misconstrued in a manner that underestimates it. The authors estimate a per-patient use cost by taking the full annual cost (purchase price divided by useful life), and apportioning to an average patient the average proportion of a year a patient utilizes the monitoring system. This is incorrect in that it results in treating the time that the system is unused as costless, as though the NHS could only cover a percentage of the purchase price equal only to the percent of the year it will be actively used. Even as a distal payer, the full cost of the system will be passed to the NHS, and the true cost per patient will be very simply the annual cost divided by annual patients. Estimating this way will raise the estimated intervention cost per patient, and every ICER reliant upon it. However, the change will not alter findings to an extent that will alter the fundamental finding that the system will be well with in cost-effectiveness thresholds.

There is also another aspect of this paper that may or may not be correct, and discussion (if not revision) is warranted. It is very noteworthy that the (adjusted) difference in point estimates of the mortality between intervention and control (as presented in Table 2 and Table S9, roughly -.001) in are actually quite small and do not match the adjusted mortality reductions discussed in the Clinical Background section and reported in reference 14 (roughly -.016, AOR ~.079). Why is this? This difference certainly has a large influence on the estimate of the resulting effectiveness inputs. Discussion and possibly revision are warranted. This difference may be underselling the cost-effectiveness of the system.

These being my main concerns with the manuscript, a handful of other areas would benefit from further attention:

- On line 49, for clarity it would help if the phrasing were "remains cost-effective at 100% 'usage' " lest some readers infer a meaning of "remains cost effective at 100% probability."

- The "deterioration" which triggers a notification should be more clearly defined in the clinical background section. I found myself turning to the underlying investigation (ref 14) to understand this. It was otherwise somewhat unclear at first read if a notification could meaningfully precede "events" and forestall them or simply more vigorously rallied clinical staff when an event was already occurring.

- On line 141 refers readers to S1 to understand how QALYs are estimated in the model. The example in S1 is well taken, However, the reader also needs to be referred to Table 2 for the sources of QALYs, OR they need to be also reported in S1. At least a sentence or two on how the authors searched for and/or selected these as their sources is also warranted. These are very key inputs.

- Regarding lifetime resource use as discussed briefly in lines 146-150, this deserves more explanation. I see estimates with sources in Table 2, and the reader should be pointed there. Moreover, if the utilized estimates are directly reported in those sources, that should be mentioned in this paragraph. If the authors did analysis with those sources to arrive at the estimates in Table 2, that analysis should be described here.

- Regarding the equations in lines 158-160, it is appreciated that the authors use as their main analysis a conservative 5-yr useful product life, and leave longer estimates to sensitivity analyses. For completeness, if training costs are a regular part of initiating use of this technology, that should also be included, or their omission mentioned as a limitation. Similarly, if maintenance costs are a regular cost of utilizing this technology, they should also be incorporated or their omission should be mentioned as a limitation.

- Regarding the footnotes in Table 1 and the related S2-5: The weighted average should mention what weights; I'm inferring "frequency weighted average," correct? Relatedly, in S5, is "activity" synonymous with "frequency"? And is S5 and input into S4? In which case I would re-order them.

- Lines 185 - 187 is the only place in the main text where discounting is discussed, and only with reference to the inpatient stays. Is discounting performed for lifetime (post-stay) costs and/or QALYs?

- the Long-term costs section should at some point refer the reader clearly to Table 2.

- Within Table 2, the "Distribution" column should also report either the relevant confidence interval or variance parameter around the point estimate in the 'point-estimate' column.

- In Table 4, I note that though Quality-adjusted life expectancy is higher in the Intervention group, raw life expectancy is reported at about .014 life years shorter. Is this accurate? Also, why is this the only row in Table 4 to not have 95% central ranges? They should be included in this row as with the others if available.

- The note in Table S9 states the adjusted probabilities are adjusted for "age, sex, gender...." I wanted to check that it's accurate that the adjustment included both sex and gender distinctly construed? Or was it just one of them?

These detailed comments notwithstanding, the authors should be commended on a solid investigation, and I would expect that with revision this paper will reach a publishable form.

6. PLOS authors have the option to publish the peer review history of their article (what does this mean?). If published, this will include your full peer review and any attached files.

Reviewer #1: **Yes: **Dr. Seyyed Mostafa Hakimzadeh

Reviewer #2: No

---

## [Author Response · Author response to Decision Letter 0]

24 Nov 2023

A rebuttal letter has been uploaded as part of the submission. We hope that this satisfies the editors and reviewers.

---

## [Decision Letter · Decision Letter 1]

12 Jan 2024

PONE-D-23-27466R1A model-based cost-utility analysis of an automated notification system for deteriorating patients on general wardsPLOS ONE

Dear Dr. Subbe,

Thank you for submitting your manuscript to PLOS ONE. After careful consideration, we feel that it has merit but does not fully meet PLOS ONE’s publication criteria as it currently stands. Therefore, we invite you to submit a revised version of the manuscript that addresses the points raised during the review process.

We look forward to receiving your revised manuscript.

Kind regards,

Dominic Luke Thorrington, PhD

Academic Editor

PLOS ONE

**Additional Editor Comments:**

One of the previous reviewers has highlighted a minor issue that needs to be dealt with, whilst a new reviewer has left several major comments on the manuscript:

 - a more detailed description of the model is required

 - some cost items in the economic model are missing, which the reviewer highlights as a serious flaw

 - additional information is required for the VITAL II study, given its importance to this manuscript

 - some additional information is needed to describe the general modelling process for the reader In addition to the bullet points above, I invite you to read the detailed feedback from both reviewers before proceeding with amendments to your manuscript.

Reviewers' comments:

Reviewer's Responses to Questions

**Comments to the Author**

1. If the authors have adequately addressed your comments raised in a previous round of review and you feel that this manuscript is now acceptable for publication, you may indicate that here to bypass the “Comments to the Author” section, enter your conflict of interest statement in the “Confidential to Editor” section, and submit your "Accept" recommendation.

Reviewer #2: (No Response)

Reviewer #3: (No Response)

2. Is the manuscript technically sound, and do the data support the conclusions?

Reviewer #2: Yes

Reviewer #3: No

3. Has the statistical analysis been performed appropriately and rigorously? 

Reviewer #2: Yes

Reviewer #3: No

4. Have the authors made all data underlying the findings in their manuscript fully available?

Reviewer #2: Yes

Reviewer #3: No

5. Is the manuscript presented in an intelligible fashion and written in standard English?

Reviewer #2: Yes

Reviewer #3: No

6. Review Comments to the Author

Reviewer #2: This revised manuscript has made excellent progress in addressing my previous comments, most of which have been thoroughly cleared up. I hold to but one sticking point on the per-patient costs as discussed in my primary comment in the original description.

The authors have now added clarity that their calculations assume an annual patient number based on operating at full ward capacity. Because this assumption is now explicit, this tact is acceptable so long as they add somewhere in the discussion or limitations sections that this assumption results in an optimistic bound for this parameter's effect on the cost effectiveness of the intervention. However, having re-checked reference 14, it is also suggested that the authors note that the wards did run at near capacity during the intervention. [To wit, 2263 patients were served in in the ~25 ward months as opposed to the expected 357/8.62*54 = 2286 patients annually across the two wards]. Alternately, the authors may re-run their analyses with a more empirical annual number of patients, but that would seem a fair amount of effort for a minor shift in precision, and I'm not requiring that so long as the optimistic effect of their 100% capacity assumption is explicitly noted.

Otherwise, this is a strong analysis that I expect to approve for publication pending this minor revision.

Reviewer #3: This is an interesting paper which reports the cost-effectiveness analysis of automated notification system for deteriorating patients on general wards in the UK. While the topic is worthy of study, the current paper does not provide sufficient detail on the methods used, I have concerns over some of the approaches taken and additional detail is required throughout. Please find some specific comments below.

Major comments:

1) A more detailed description of the model is required, including an improved model schematic. For example, it is unclear how repeat events have been addressed in the model.

2) The study does not appear to include any costs assocaited with responding to a notification system- this appears to be a significant oversight.

3) A more detailed description of the VITAL II Study is required (e.g. patients included, types of ward etc)

4) More details on cost-effectiveness methods need to be provided for a clinical audience (e.g. what is a QALY, a threshold, discounting, cost-effectiveness decision rules etc)

further comments

1) The methods section of abstract should be expanded- for example, including scenario analyses

2) The authors should complete a CHEERS checklist to ensure that the economic evaluation is appropriately reported

3) WHow is NEWS measured? (Note first time it is used as just an acronym with this not defined until line 131)

4) Why did you restict patients in VITAL to those with NEWS of 15?

5) For what purpose was a review of economic evaluations conducted? No real mention of how if at all these were used to inform this study.

6) Need to provide details on the lifetime costs used (I don't see these reported in Table 1 as stated)

7) I don't think providing the distributions for the PSA in Table 1 is particularly useful here (particularly the numbers used). Would be better to report something on the parameter uncertainty (e.g. standard error)

8) Following on- why use arbitrary level of parameter uncertainty of 0.2 of mean? This seems unnecessaary for many of the parameters used where you would be able to estimate the actual uncertainty

9) Annuitization should be used to calculate the cost of the product per patient

10) More justication is required for the 100% ward capacity assumption

11

7. PLOS authors have the option to publish the peer review history of their article (what does this mean?). If published, this will include your full peer review and any attached files.

Reviewer #2: No

Reviewer #3: No

---

## [Author Response · Author response to Decision Letter 1]

25 Feb 2024

Dr Christian P Subbe, DM, FRCP Menai Bridge, the 25th of February 2024

21 Menai Quays

Menai Bridge

LL59 5DB

To the Academic Editor

Dominic Luke Thorrington, PhD

PLOS ONE

Re: PONE-D-23-27466

A model-based cost-utility analysis of an automated notification system for deteriorating patients on general wards 

Many thanks for the comments from the reviewers and invitation to submit a revised version of the manuscript. In response, we have revised the manuscript, providing a more detailed description of the model, greater clarity and justification of costs, and further description of the VITAL II study. 

The following documents have been uploaded. 

• A rebuttal letter that responds to each point raised by the academic editor and reviewer(s). You should upload this letter as a separate file labelled 'Response to Reviewers'. 

• A marked-up copy of your manuscript that highlights changes made to the original version. You should upload this as a separate file labelled 'Revised Manuscript with Track Changes'. 

• An unmarked version of your revised paper without tracked changes. You should upload this as a separate file labelled 'Manuscript'.

There are no changes to the financial disclosure. 

Line references in the rebuttal letter correspond to ‘Revised Manuscript with Track Changes’.

Review Comments to the Author

Reviewer #2: This revised manuscript has made excellent progress in addressing my previous comments, most of which have been thoroughly cleared up. I hold to but one sticking point on the per-patient costs as discussed in my primary comment in the original description.

The authors have now added clarity that their calculations assume an annual patient number based on operating at full ward capacity. Because this assumption is now explicit, this tact is acceptable so long as they add somewhere in the discussion or limitations sections that this assumption results in an optimistic bound for this parameter's effect on the cost effectiveness of the intervention. However, having re-checked reference 14, it is also suggested that the authors note that the wards did run at near capacity during the intervention. [To wit, 2263 patients were served in in the ~25 ward months as opposed to the expected 357/8.62*54 = 2286 patients annually across the two wards]. Alternately, the authors may re-run their analyses with a more empirical annual number of patients, but that would seem a fair amount of effort for a minor shift in precision, and I'm not requiring that so long as the optimistic effect of their 100% capacity assumption is explicitly noted.

Otherwise, this is a strong analysis that I expect to approve for publication pending this minor revision.

Response by the authors: We thank Reviewer #2 for providing feedback on the revisions and note the comment that this is a strong analysis.

We can confirm that the study wards (as wards in most hospitals of the National Health Service) run at very close to 100 percent of capacity. It is usual practice in NHS hospitals to fill ward to capacity to create space at ‘the front door’ for assessment of new patients.

As requested by reviewer #2, we have noted this as a limitation (lines 425-427):

 “Finally, the assumption of 100% ward capacity, may be judged to be an optimistic bound, however, it is usual practice in NHS hospitals to fill ward to capacity to create space at ‘the front door’ for assessment of new patients.”

Reviewer #3: This is an interesting paper which reports the cost-effectiveness analysis of automated notification system for deteriorating patients on general wards in the UK. While the topic is worthy of study, the current paper does not provide sufficient detail on the methods used, I have concerns over some of the approaches taken and additional detail is required throughout. Please find some specific comments below.

Major comments:

1) A more detailed description of the model is required, including an improved model schematic. For example, it is unclear how repeat events have been addressed in the model.

Response by the authors: We had edited the model overview section to more clearly explain that the first part of the model is a patient level analysis, based on observed patient data, that is then extrapolated to a life-time horizon (lines 108-111) to; and, to clarify why literature reviews were required (line 138-140). 

Lines 154-162 detail how repeat events have been addressed in the model:

“During the inpatient episode the model accounted for multiple events per patient. Health states at discharge were defined by the principal serious adverse event during inpatient episode. Where patients experienced multiple events the event with the worst health state was assumed at discharge.”

Probability of event was calculated using negative binomial regression to allow for baseline differences in age, gender, ward, and NEWS score on admission. This detail has been added to the analysis section of the revised manuscript (lines 232-241).

For clarity we have revised the Figure 1 to indicate the cost-utility model more clearly (Fig 1_revised). 

2) The study does not appear to include any costs associated with responding to a notification system- this appears to be a significant oversight.

Response by the authors: The rapid response team is a fixed cost that is independent of frequency of alerts. As stated in the manuscript, hospital policy stipulates this remains the same in both arms of the model: automated advisory vital signs monitoring system OR non-connected spot-check monitors (control). 

Our analysis reflects this policy. The opportunity cost of automated versus human, however, is reported in the Discussion (lines 417-420): 

The analysis did not account for the opportunity cost of automated versus human monitoring, whist this replicates policy (staffing levels are required to remain constant), time spent on monitoring represents resource that could be redistributed to other elements of care.

3) A more detailed description of the VITAL II Study is required (e.g. patients included, types of ward etc)

Response by the authors: This is a secondary analysis of the VITAL II study and details have been published in the reference open access publication. We have however added in the method section additional details (lines 116-121):

The model captures all events during the inpatient stay based on data obtained from the Vital II study. Patients were admitted to the study wards following a short period of assessment and completion of admission documentation in the Acute Medical Unit of the hospital in line with usual practice in the NHS. One of the wards specialised in Respiratory and one in Gastroenterological conditions but both wards took patients with other conditions. … …. 

4) More details on cost-effectiveness methods need to be provided for a clinical audience (e.g. what is a QALY, a threshold, discounting, cost-effectiveness decision rules etc)

Response by the authors: It is our understanding that original Health Economics research is within the scope, and it would be unusual to find 101 guides to other disciplines in other papers. We do, however, appreciate the opportunity to refer readers to general texts, to gain a more comprehensive overview of concepts and as such, we have provided a footnote (line 95) for the editor to consider:

Economic evaluation provides a framework in which to assess the costs and effects of alternative interventions, such as automated monitoring compared to standard care. For a comprehensive overview of concepts and methods, readers should refer to general texts, such as: Morris, Stephen, et al. Economic analysis in healthcare. John Wiley & Sons, 2012.

Furthermore, we have reviewed the health economics terms used in the manuscript and provided additional information to support the reader e.g., Defined QALY (line 111) and on line 108 we have included the associated ratio alongside the description of the analysis framework: “…. cost-effectiveness analysis (cost per evet avoided).”

Further comments

1) The methods section of abstract should be expanded- for example, including scenario analyses

Response by the authors: We have revised the method section of the abstract to include reference to sensitivity analysis (line 37-38).

2) The authors should complete a CHEERS checklist to ensure that the economic evaluation is appropriately reported

Response by the authors: The CHEERS checklist was included with the original submission, and we regret that Reviewer #3 did not have access to this document. We have included an updated version with our resubmission, with page references corresponding to the unmarked version of the revised paper labelled 'Manuscript’.

3) WHow is NEWS measured? (Note first time it is used as just an acronym with this not defined until line 131)

Response by the authors: We have revised the manuscript to define National Early Warning Score (NEWS) (14) on first use (line 81) and updated the references accordingly (previously reference 15). We have also included a new footnote for those unfamiliar with the NEWS (line 81):

 “The National Early Warning Score is a score that summarises abnormalities in vital signs such as blood pressure, heart rate, temperature through a point system ranging from zero (all parameters normal) to 20 (all parameters maximally abnormal)”

4) Why did you restict patients in VITAL to those with NEWS of 15?

Response by the authors: The present analysis was restricted to patients who had a recorded NEWS score on admission to be able to risk stratify the impact of the intervention. We did not restrict to NEWS of 15. We are assuming the reviewer cites “(restricted to cases with National Early Warning Score (NEWS)(15) score on admission)” – where (15) is the reference for the NEWS score.

 To avoid confusion, we have revised the manuscript (line 138-140) to explicitly state ‘restricted to complete NEWS score on admission; and referenced the National Early Warning Score at first use (line 81; see response to (3) above). 

5) For what purpose was a review of economic evaluations conducted? No real mention of how if at all these were used to inform this study.

Response by the authors: The review of economic evaluations was conducted to ascertain the life-time parameters for the cost-utility analysis. We have revised the manuscript to state this explicitly (line 140). 

… and purposive reviews of the literature to obtain long-term estimates of costs and outcomes, in line with standard methodology for populating economic models (16).

6) Need to provide details on the lifetime costs used (I don't see these reported in Table 1 as stated)

Response by the authors: Lifetime costs are detailed (and referenced) in Table 1 (line 173) table rows 28-36. We have added sub-headings to this table to aid the reader. 

7) I don't think providing the distributions for the PSA in Table 1 is particularly useful here (particularly the numbers used). Would be better to report something on the parameter uncertainty (e.g. standard error)

8) Following on- why use arbitrary level of parameter uncertainty of 0.2 of mean? This seems unnecessaary for many of the parameters used where you would be able to estimate the actual uncertainty

Response by the authors: The manuscript follows standard reporting for health economic models. Providing the distributions for the PSA in Table 1 provides transparency and allows for replication of analysis. The 95% central range for parameters in the patient level analysis (inpatient costs) have been added to Table 1 (line 173; rows 26-27) . Parameter uncertainty of 0.2 of the mean was applied to secondary estimates obtained from published models, that were adjusted for the age and sex match of the population. We have performed a sensitivity analysis of the standard deviation using 0.1 (half of the base case assumption) and 0.4 (double the base case assumption) and conclude that the probability of cost-effectiveness appears robust to the assumptions around the variance. 

Parameter 

Standard deviation £20,000 £30,000

0.1 0.8157 0.8011

0.2 0.8082 0.7889

0.4 0.7918 0.7769

 This has been reported within the narrative of revised manuscript, methods (lines 270) and results (lines 358-359).

9) Annuitization should be used to calculate the cost of the product per patient

Response by the authors: The cost of the product per patient is presented in constant, real terms, as specified on lines 187. This accounts for the product-life (5-years) and assumes a single payment purchase price at zero interest rate (using straight line depreciation). On reflection of the reviewer’s comment calculation of equivalent annual cost for both intervention and control, with a product life of 5-years and discount rate of 3.5% (annuity factor 4.515) has been performed. This adjusts the incremental cost marginally; by £0.56, which remains below the threshold at which the intervention remains cost-saving (up to £12.08). The costing method has been clarified in the revised manuscript (line 192 and line 263-4) and product cost per patient when with annuity factor 4.515 has been inserted as a footnote to table 3 (line 322-3).

10) More justication is required for the 100% ward capacity assumption

We have noted this assumption as a limitation (lines 425-427):

“Finally, the assumption of 100% ward capacity, may be judged to be an optimistic bound, however, it is usual practice in NHS hospitals to fill ward to capacity to create space at ‘the front door’ for assessment of new patients.”

Note to editor: Our response here is consistent with the recommendation of reviewer 2: 

Reviewer #2: Alternately, the authors may re-run their analyses with a more empirical annual number of patients, but that would seem a fair amount of effort for a minor shift in precision, and I'm not requiring that so long as the optimistic effect of their 100% capacity assumption is explicitly noted.

Otherwise, this is a strong analysis that I expect to approve for publication pending this minor revision.

Many thanks for the kind comment. We are looking forward to the feedback from the editor. 

Yours truly,

Dr. Christian P Subbe, DM, FRCP

Consultant in Acute and Critical Care Medicine

Senior Clinical Lecturer, School of Medical Sciences

Bangor University, Wales, United Kingdom

---

## [Decision Letter · Decision Letter 2]

20 Mar 2024

A model-based cost-utility analysis of an automated notification system for deteriorating patients on general wards

PONE-D-23-27466R2

Dear Dr. Subbe,

We’re pleased to inform you that your manuscript has been judged scientifically suitable for publication and will be formally accepted for publication once it meets all outstanding technical requirements.

An invoice for payment will follow shortly after the formal acceptance. To ensure an efficient process, please log into Editorial Manager at Editorial Manager® , click the 'Update My Information' link at the top of the page, and double check that your user information is up-to-date. If you have any billing related questions, please contact our Author Billing department directly at authorbilling@plos.org.

Kind regards,

Dominic Luke Thorrington, PhD

Academic Editor

PLOS ONE

Additional Editor Comments (optional):

Thank you for responding to the reviewers' recommendations and comments on the first revision of the manuscript. The manuscript is now much improved.

Reviewers' comments:

Reviewer's Responses to Questions

**Comments to the Author**

1. If the authors have adequately addressed your comments raised in a previous round of review and you feel that this manuscript is now acceptable for publication, you may indicate that here to bypass the “Comments to the Author” section, enter your conflict of interest statement in the “Confidential to Editor” section, and submit your "Accept" recommendation.

Reviewer #3: All comments have been addressed

2. Is the manuscript technically sound, and do the data support the conclusions?

Reviewer #3: Yes

3. Has the statistical analysis been performed appropriately and rigorously? 

Reviewer #3: Yes

4. Have the authors made all data underlying the findings in their manuscript fully available?

Reviewer #3: Yes

5. Is the manuscript presented in an intelligible fashion and written in standard English?

Reviewer #3: Yes

6. Review Comments to the Author

Reviewer #3: The authors have responded in a satisfactory manner to all of my comments. I have no further comments to add/

7. PLOS authors have the option to publish the peer review history of their article (what does this mean?). If published, this will include your full peer review and any attached files.

Reviewer #3: No
